# Diffusion Transformer for Adaptive Text-to-Speech

*Haolin Chen[1,2], Philip N. Garner[1]*

[1]Idiap Research Institute, Martigny, Switzerland
[2]École Polytechnique Fédérale de Lausanne, Switzerland

hchen@idiap.ch, pgarner@idiap.ch

## Abstract

Given the success of diffusion in synthesizing realistic speech, we investigate how diffusion can be included in adaptive text-to-speech systems. Inspired by the adaptable layer norm modules for Transformer, we adapt a new backbone of diffusion models, Diffusion Transformer, for acoustic modeling. Specifically, the adaptive layer norm in the architecture is used to condition the diffusion network on text representations, which further enables parameter-efficient adaptation. We show the new architecture to be a faster alternative to its convolutional counterpart for general text-to-speech, while demonstrating a clear advantage on naturalness and similarity over the Transformer for few-shot and few-parameter adaptation. In the zero-shot scenario, while the new backbone is a decent alternative, the main benefit of such an architecture is to enable high-quality parameter-efficient adaptation when finetuning is performed.

**Index Terms**: speech synthesis, adaptive TTS, diffusion transformer, adaptive layer norm

## 1. Introduction

Adaptive text-to-speech (TTS) [1, 2, 3, 4] aims to synthesize personalized voices of target speakers or speaking styles. In the typical scenario of adaptive TTS, a source acoustic model pre-trained on a large multi-speaker corpus is adapted with limited data of the target to synthesize the desired voice. In general, adaptive TTS systems should be well generalizable and adaptable to various speaker traits and acoustic conditions with as few data as possible. Meanwhile, the adapted voice should be of high quality and naturalness, in terms of which deep generative models [5, 6, 7] have demonstrated their superiority over previous solutions. In particular, the more recent diffusion models [7, 8, 9] have dominated in terms of quality and naturalness.

While the generalizability and adaptability have been the most important properties of adaptive TTS systems and in many cases interrelated, they can be attributed to different parts of the model or algorithm design. On the one hand, the techniques that improve the ability to generalize to various features in speech signals can be categorized into 1) employing reference encoders to generate representations of the desired attribute of speech on various semantic levels [3, 4, 10], which are normally plugged in before the decoder; 2) learning algorithms that help factorize such representations into expressive components [1, 2, 11], which are usually combined with reference encoders; and 3) ad-hoc designs of the model structure that control desired features [2, 3, 12], which are more model-specific. On the other hand, adaptability, while partly overlapping with the former, emphasizes more the application itself, including considerations of few-data [3, 13], few-parameter [3] and zero-shot [4, 14] scenarios. However, no matter in which concept, there is a clear distinction between generic techniques that fit different backbones, such as reference encoders, and ones with ad-hoc architectural designs of the network. The latter are more associated with the adaptability of the model, especially in few-data and parameter-efficient settings. Furthermore, when combined with generic adaptation techniques, such architectures will enable both compute-efficient zero-shot adaptation, and high-quality adaptation when finetuning is performed.

In general, we are interested in integrating adaptable components into diffusion-based acoustic models that add extra adaptability on top of their high-quality synthesis. Despite diffusion models having been well studied for general acoustic modeling, few works have explored them for adaptive TTS systems. Guided-TTS 2 [13] utilizes diffusion with classifier guidance to adapt to diverse voices, while lacking parameter efficiency since the whole decoder needs finetuning during adaptation. In Grad-StyleSpeech [15], the diffusion mostly works as a post-net that refines the output of an adaptive Transformer decoder, and the researchers only tested adapting the whole diffusion post-net in the few-shot setting. Another study [16] shows the convolutional diffusion decoder can be adapted using conditional layer normalization, however, it must be used with adaptive Transformer layers to achieve usable adaptation quality. Our search for solutions focuses on the architecture design of the diffusion backbone. Such a design will not only facilitate parameter-efficient adaptation during finetuning, but also has the potential to be combined with reference encoders to improve the network's generalizability.

In this paper, we propose to adapt a novel backbone of diffusion models, Diffusion Transformer (DiT) [17], for adaptive TTS. Inspired by the recent innovation in image synthesis and the effectiveness of conditional layer norm [2, 3, 14] in the Transformer network, we adapt the DiT's adaptive layer norm to receive a sequence as condition instead of the class embedding, which is suitable for TTS tasks. Through a series of experiments, we demonstrate that 1) for general TTS tasks, the DiT can serve as a substitute backbone for present diffusion decoders in the acoustic model, yielding comparable performance to current designs while providing faster synthesis; 2) for few-shot adaptation, the benefits of the DiT include its capability to perform parameter-efficient adaptation, and its superiority in speech quality and similarity over previous Transformer-based solutions; 3) when based on zero-shot adaptation solutions, the DiT can efficiently achieve high-quality adaptation when finetuning is necessary. Audio samples are available at https://recherchetts.github.io/dit/.

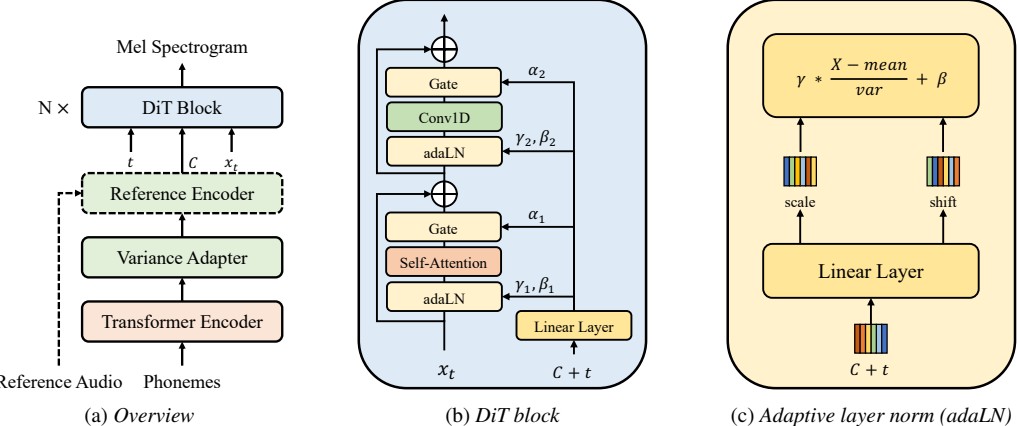

(a) *Overview*          (b) *DiT block*          (c) *Adaptive layer norm (adaLN)*

Figure 1: *The architecture of the DiT-based acoustic model. The reference encoder only exists in adaptive TTS systems.*

## 2. Diffusion Transformer for TTS

Like other deep generative model-based solutions, a typical diffusion-based acoustic model comprises a Transformer text encoder and a diffusion decoder. Essentially, diffusion models generate high-quality and natural samples by denoising a sample from a prior distribution to real data through a diffusion process. In most cases, the learning problem of diffusion can be expressed as learning a denoiser network that predicts the noise in each diffusion step, while other parameterization forms of the denoiser also exist.

### 2.1. Architecture

In principle, the denoiser network takes the sample from the previous step as input to predict the noise in the reverse diffusion process while being conditioned on text representations $C$ and the step embedding $t$. The network design enjoys flexibility as long as its output has the same dimension as the input. The prevalent architectures of the denoiser network in acoustic models include the bidirectional dilated convolutional network (CNN) [8, 7, 18], also referred to as the non-causal WaveNet [19], and the U-Net [9, 13]. The former is best known for the inductive bias of audio signals and also commonly used in variational autoencoders [20] and flow models [5, 6, 21], while the U-Net [22] is a generic network that originates from image processing.

Recently, Peebles and Xie [17] proposed a new class of diffusion models based on the Transformer architecture, namely Diffusion Transformer (DiT), which was shown to outperform U-Net backbones and inherit the scalability, robustness and efficiency of the Transformer model class. As is depicted in Figure 1b, the DiT blocks receive the sample from the last step as input, perform the common transformations of the Transformer and generate the output. The innovation of DiT lies in the way conditions are injected into the network: the standard layer norm modules in the Transformer blocks are replaced with adaptive layer norm (adaLN), so that the dimension-wise scale and shift parameters $\gamma$ and $\beta$ can be regressed from the sum of the class embedding $c$ and the step embedding $t$ through a linear layer. In addition to adaLN, the authors further propose to zero-initiate the final adaptive layer norm in each block to accelerate convergence, and also regress scaling parameters $\alpha$ that are placed before any residual connections within the DiT block. This is referred to as adaLN-Zero. The authors demon-

strate that adaLN-Zero achieves the best performance and adds the least computation cost to the model compared to introducing conditions by in-context learning and cross-attention.

The original DiT was tested on image synthesis tasks, in which only the class embedding controls the content to be generated. To adapt it for TTS, we make the adaLN-Zero accept a sequence of encoded text representations. In actuality, the implementation of adaLN-Zero requires no modification whatsoever. The novelty lies in the fact that the regression of all scale and shift parameters is now performed on the sum of the text representation matrix $C$ and the step embedding $t$, generating the necessary scale and shift parameters for each vector in the input sequence, as is shown in Figure 1c. Note that the size of the text representation matrix matches that of the hidden representations in the DiT block, since they are both expanded to the length of the mel spectrogram using phoneme durations. Therefore, instead of the same scale and shift vectors applied on the entire input sequence in the affine transform of the layer norm, a sequence of such vectors with the same dimension as the input is applied. This allows the adaptive layer norm to modulate the input sequence using the text representations without adding any computation cost compared to the original adaLN.

### 2.2. Generator-based diffusion

The common parameterization method of diffusion is to let the neural network be a noise predictor. It originates from the reverse diffusion process:

$$p_\theta\left(\mathbf{x}_{t-1} \mid \mathbf{x}_t\right) = \mathcal{N}\left(\mathbf{x}_{t-1}; \boldsymbol{\mu}_\theta\left(\mathbf{x}_t, t\right), \boldsymbol{\Sigma}_\theta\left(\mathbf{x}_t, t\right)\right) \quad (1)$$

where the reverse transition probability $p_\theta\left(\mathbf{x}_{t-1} \mid \mathbf{x}_t\right)$ is parameterized by a neural network $\theta$. By setting $\boldsymbol{\Sigma}_\theta\left(\mathbf{x}_t, t\right)$ to a constant and reparameterizing $\mathbf{x}_0 = \frac{1}{\sqrt{\bar{\alpha}_t}}\left(\mathbf{x}_t - \sqrt{1-\bar{\alpha}_t}\boldsymbol{\epsilon}\right)$ which is derived from the noise adding function of the forward process: $q(\mathbf{x}_t|\mathbf{x}_0) = \mathcal{N}(\mathbf{x}_t; \sqrt{\bar{\alpha}_t}\mathbf{x}_0, (1-\bar{\alpha}_t)\mathbf{I})$, the problem of learning $\boldsymbol{\mu}_\theta$ can be converted to estimating the Gaussian noise $\epsilon$, resulting in the following simplified loss function:

$$L_{\text{simple}}\left(\theta\right) = \mathbb{E}_{t,\mathbf{x}_0,\epsilon}\left[\|\epsilon - \epsilon_\theta\left(\mathbf{x}_t, t\right)\|^2\right] \quad (2)$$

With the diffusion in this form, it usually requires hundreds to thousands of denoising steps to ensure high-quality synthesis.

An alternative way to parameterize the denoiser is to make it directly predict the clean data in each denoising step. Specifically, the neural network $f_\theta(\mathbf{x}_t, t)$ that outputs $\mathbf{x}_0$ given $\mathbf{x}_t$ now

models the distribution $p_\theta(\mathbf{x}_0 \mid \mathbf{x}_t)$. Next, $\mathbf{x}_{t-1}$ is sampled using the posterior distribution:

$$q\left(\mathbf{x}_{t-1} \mid \mathbf{x}_t, \mathbf{x}_0\right) = N\left(\mathbf{x}_{t-1}; \tilde{\mu}_t\left(\mathbf{x}_t, \mathbf{x}_0\right), \tilde{\beta}_t \mathbf{I}\right)$$

$$\tilde{\mu}_t\left(\mathbf{x}_t, \mathbf{x}_0\right) = \frac{\sqrt{\bar{\alpha}_{t-1}}\beta_t}{1-\bar{\alpha}_t}\mathbf{x}_0 + \frac{\sqrt{\alpha_t}\left(1-\bar{\alpha}_{t-1}\right)}{1-\bar{\alpha}_t}\mathbf{x}_t, \quad (3)$$

$$\tilde{\beta}_t = \frac{1-\bar{\alpha}_{t-1}}{1-\bar{\alpha}_t}\beta_t.$$

The rest of the inference process remains the same. The loss is then defined in the data space:

$$L_{\text{simple}}^{\text{gen}}(\theta) = \mathbb{E}_{t,\mathbf{x}_0}\left[\|\mathbf{x}_0 - f_\theta\left(\mathbf{x}_t, t\right)\|^2\right] \quad (4)$$

This parameterization method is sometimes referred to as the generator-based method [23, 18]. Some recent works [18, 24] utilize this method to enable fast synthesis for diffusion-based acoustic models. Huang et al. [18] compared the generator-based method with the conventional denoising-based method with varying diffusion steps and found that the former achieved the highest quality in all settings. To accelerate inference while maintaining high synthesis quality, we adopt the generator-based method in our model.

### 2.3. Comparison with baseline

We first test our model on basic TTS tasks and compare the DiT architecture with the prevalent non-causal WaveNet. We would expect the DiT to perform identically to the baseline in terms of speech quality.

**Implementation details.** Both models consist of a 4-layer Transformer phoneme encoder with a hidden size of 256, a variance adapter same as the one in FastSpeech 2 [25], and a diffusion decoder. The DiT network is configured as 4-layer with a hidden size of 256 and 2 attention heads, which is the same as a commonly-used Transformer decoder, while the WaveNet network is set to 20-layer with 256 hidden size. Our implementation is based on the open-source software [1] [2] of related models. The numbers of parameters of the WaveNet-based model and the DiT-based model are 30.50M and 28.83M, respectively.

**Data.** We train the models on the single speaker corpus LJSpeech [26]. Two sets of 500 utterances are selected as the validation and test set, while the rest are used as training set. All data are preprocessed following the practice in FastSpeech 2, with a sampling rate of 22,050 Hz.

**Training and inference.** The models are trained on one NVIDIA RTX3090 GPU using a batch size of 40,000 speech frames, with the "rsqrt" (reciprocal of the square root) scheduler, 4,000 warm-up steps, and a learning rate factor of 2. For the diffusion process, a beta schedule of 16 steps is used for both training and inference. A HiFi-GAN [27] vocoder trained on LJSpeech is used to synthesize waveforms. The inference is performed on the same hardware.

**Evaluation.** For objective evaluation, we utilize the Speech-Brain [28] toolkit to run speaker verification and speech recognition [3] on the entire test set. The averaged speaker cosine similarity (SPK) and character error rate (CER) are calculated as indicators of how well the model captures the speaker identity and the intelligibility of synthesized samples. For subjective evaluation, we recruited 20 native raters on Prolific [4] crowdsourcing platform to rate the overall quality and naturalness of

---

[1] NATSpeech: https://github.com/NATSpeech/NATSpeech
[2] DiT: https://github.com/facebookresearch/DiT
[3] spkrec-ecapa-voxceleb; asr-wav2vec2-librispeech
[4] https://www.prolific.co

Table 1: *The MOS scores with 95% confidence, SPK and CER scores on LJSpeech, and real-time factors.*

| Arch. | MOS (↑) | SPK (↑) | CER (↓) | RTF (↓) |
|-------|---------|---------|---------|---------|
| *Vocoder* | 4.35 ± 0.10 | 0.983 | 1.83% | - |
| *WaveNet* | 4.06 ± 0.10 | 0.790 | 2.41% | 0.021 |
| *DiT* | 4.01 ± 0.10 | 0.784 | 2.38% | 0.012 |

randomly selected 20 samples from the test set using the P.808 toolkit [29]. We also calculate the real-time factor (RTF) of the two models that reflects the synthesis speed, which is conducted when synthesizing around 200 paragraphs.

### 2.4. Results

All test results are listed in Table 1. The subjective test results show the DiT architecture has a gap of only 0.05 compared to the non-causal WaveNet within the 95% confidence interval of 0.10 which, consistent with our expectation, suggests the DiT offers a similar synthesis quality to the prevalent architecture. This is also reflected on the two objective test scores, which only demonstrate minor difference between the two architectures.

The RTFs indicate that the model with a DiT backbone is overall 70% faster than the one with a WaveNet backbone, using the model configuration above. By breaking down the time cost into different components, we found that the 4-layer DiT-based decoder has around 2.4 times the speed of a 20-layer WaveNet-based decoder.

Overall, the results of the basic TTS task demonstrate that the DiT is a faster alternative of the diffusion backbone to the non-causal WaveNet, which also shows a slight advantage on the model size. This is perhaps not persuasive enough for switching the diffusion backbone, however, the merit of DiT lies in its ability to be adapted efficiently, which will be elaborated in the next section.

## 3. Adaptive DiT

### 3.1. Method

In Transformer [30], the layer norm [31] helps reduce the variance of the hidden representations after the attention and feed-forward transformation to stabilize and speed up training. Previous works [2, 3, 14] have found that the layer norm in Transformer can greatly influence the hidden activation and the final prediction with the learnable scale and shift parameters. Furthermore, these parameters can be regressed from the speaker or style representation, e.g. the speaker embedding, through a small neural network, which can be finetuned during adaptation. The method significantly reduces the number of parameters to be adapted for each new speaker or style, while maintaining high-quality synthesis.

As for DiT, the architecture unification enables us to apply the same method to the adaptive layer norm. Inherently, the adaLN receives all the conditional input to the decoder, including the speaker embedding and possibly embeddings from reference encoders. The nature cancels the requirement for any additional input to the decoder.

In the following experiments, we compare our adaptive DiT model with AdaSpeech, a Transformer-based solution with conditional layer norm. Given the diffusion's superiority in

Table 2: *The subjective and objective test results of few-shot adaptation experiments.*

| Dataset | | VCTK | | | | LibriTTS | |
|---|---|---|---|---|---|---|---|
| Metric | #Params | MOS ($\uparrow$) | SMOS ($\uparrow$) | SPK ($\uparrow$) | CER ($\downarrow$) | SPK ($\uparrow$) | CER ($\downarrow$) |
| *Vocoder* | - | 4.37 ± 0.08 | - | 0.955 | 3.16% | 0.929 | 2.61% |
| *AdaSpeech* | 1.184M | 2.76 ± 0.08 | 2.86 ± 0.10 | 0.505 | 3.12% | 0.508 | 3.77% |
| *DiT* | 1.711M | 3.77 ± 0.09 | 3.94 ± 0.10 | 0.570 | 2.50% | 0.582 | 3.46% |

high-quality synthesis, we would expect the DiT to offer better speech quality and speaker similarity compared to the baseline.

## 3.2. Experimental setup

**Implementation details.** We implement necessary components to construct AdaSpeech using the same TTS framework as before, including the phoneme- and utterance-level encoders in the acoustic condition modeling module and the conditional layer norm in the Transformer decoder layers. We use the same acoustic condition modeling module as AdaSpeech, thus the only difference between the DiT-based model and AdaSpeech is the decoder architecture. The model configuration of AdaSpeech follows the official settings, while the DiT follows the previous configuration.

**Data.** All models are pretrained on the two clean subsets `train-clean-360` and `train-clean-100` of the multi-speaker LibriTTS dataset [32], with a total of 1151 speakers and 245 hours. For adaptation, we use LibriTTS and the multi-speaker corpus VCTK [33] to test the in-domain and out-of-domain adaptation performances. For LibriTTS, we select 10 speakers from the `test-clean` subset, and 10 random utterances for each speaker as test set. For VCTK, 11 speakers (7 females and 4 males) with different accents are selected following [4], while for each speaker 10 utterances with the same spoken content across all speakers are selected as test set.

**Training, adaptation, and inference.** Following AdaSpeech, all models are trained in two stages in which the numbers of steps are 60,000 and 40,000 respectively, on the same hardware as before. The batch size is set to 50,000 speech frames for AdaSpeech and 40,000 for the DiT-based model. Other configurations follow the official or previous settings unless otherwise stated. During adaptation, only the speaker embedding and the layer norm modules are finetuned using 10 random utterances of the target speaker for 2,000 steps using a fixed learning rate of $2 \times 10^{-4}$. A HiFi-GAN vocoder trained on VCTK is used to synthesize waveforms.

**Evaluation.** Subjective tests are carried for the more challenging LibriTTS to VCTK out-of-domain adaptation task. The same 20 native raters are involved in the subjective test to rate the MOS for naturalness and the SMOS (Similarity MOS) for speaker similarity of 22 speaker-balanced samples from the VCTK test set generated by each system. The reference of each utterance given in the subjective test is the vocoder synthesized sample of the utterance. The objective SPK and CER scores are calculated on the entire test sets of both VCTK and LibriTTS. We calculate the number of parameters to be finetuned for each model.

## 3.3. Results and analyses

The subjective and objective test results are shown in Table 2. In the out-of-domain adaptation task, subjective test results demonstrate a clear improvement of both naturalness and

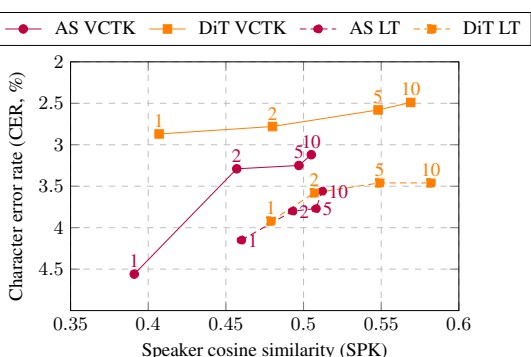

Figure 2: *The speaker cosine similarity (SPK) and character error rate (CER) of varying adaptation data. The number of utterances used for adaptation is labeled on each data point. AS: AdaSpeech, LT: LibriTTS.*

speaker similarity by the DiT decoder compared to the Transformer. In objective evaluation, the DiT achieves a higher speaker similarity score and a lower character error rate, which indicates the DiT is able to generate more intelligible speech with a voice more similar to the reference. In the in-domain adaptation task, the DiT results in a higher speaker similarity score, while AdaSpeech does not improve much. The DiT has approximately 50% more parameters finetuned compared to the Transformer, due to the extra scaling parameters $\alpha$.

We further study the naturalness and speaker similarity with varying amount of adaptation data on VCTK and LibriTTS, and conduct objective tests. As is shown in Figure 2, with increasing number of utterances used for adaptation, the speaker similarity and intelligibility continue to improve for all models and on both datasets. Overall, the DiT outperforms the Transformer in both metrics under all settings, and the difference between the two models becomes larger when the more utterances are available.

It is worth noting that, during our test listening of the adapted samples, we found AdaSpeech is more prone to the noise in the training data than the DiT, resulting in the adapted samples being more noisy. This is likely due to the low-quality samples in the `train-clean-360` subset, since adapting an AdaSpeech trained on VCTK results in a cleaner voice. Nonetheless, this phenomenon suggests the DiT is more robust against noises, which can be explained with the diffusion's denoising nature.

## 3.4. Zero-shot adaptation

Previous experiments have demonstrated that the DiT is able to generate a more high-quality voice with better similarity to the target when adapted compared to the Transformer. Although

Table 3: *The objective test results of zero-shot adaptation.*

| Arch. | AdaSpeech | | GenerSpeech | |
|---|---|---|---|---|
| Metric | SPK ($\uparrow$) | CER ($\downarrow$) | SPK ($\uparrow$) | CER ($\downarrow$) |
| *Vocoder* | 0.955 | 3.16% | 0.955 | 3.16% |
| *Transformer* | 0.107 | 2.66% | 0.292 | 6.90% |
| *DiT* | 0.132 | 2.34% | 0.299 | 4.43% |
| *WaveNet* | 0.134 | 2.20% | 0.307 | 4.06% |

we mainly focus on few-shot adaptation tasks, we are still interested to see how the architecture performs in the zero-shot setting. We also take the chance to demystify what part of the model architecture contributes the most to the generalizability of the model.

We first test the Transformer decoder, the DiT decoder, and the non-causal WaveNet-based diffusion decoder on top of the acoustic condition modeling module (the reference encoding) of AdaSpeech. All three models are trained on LibriTTS using the recipe described in Section 3.3. For inference, we randomly select one utterance from the target speaker in the VCTK test set. The objective test results are shown in Table 3. It can be observed that the DiT- and the WaveNet-based diffusion decoders bring similar slight improvements to the speaker similarity compared to the Transformer decoder, although all scores are significantly lower than few-shot adaptation. The WaveNet-based diffusion decoder seems to yield better intelligibility than DiT, however both diffusion decoders outperform the Transformer.

We further base the two diffusion decoders on a state-of-the-art zero-shot solution, GenerSpeech [10], and its official implementation [5]. All models share the same official training recipe. Note that in GenerSpeech, a flow-based post-net is used on top of the Transformer decoder to refine the output. We found the 4-layer DiT in this setting is difficult to converge, hence we use a 6-layer one instead. This time the diffusion does not show much improvement on the speaker similarity anymore compared to the Transformer. However, the two diffusion-based models yield notably higher intelligibility which is reflected on the CER, with the WaveNet backbone slight better than the DiT.

Overall, the results suggest that despite the diffusion providing slightly better speaker similarity, the bulk of generalizability lies in the reference encoding part of one adaptive system. Under these certain architectures of the acoustic model, the main benefit of a diffusion decoder in a zero-shot adaptive system is the higher-quality synthesis, rather than better similarity. In comparison with few-shot adaptation, the results also demonstrate the necessity of finetuning to achieve high similarity. On the choice of backbone architecture in the zero-shot setting, the WaveNet seems to slightly outperform the DiT. However, as is discussed above, the adaptive layer norm in the DiT backbone enables the model to be adapted efficiently when finetuning is performed, while the DiT is still a decent alternative to the prevalent non-causal WaveNet in zero-shot usage.

## 4. Conclusions

In this paper, we proposed to utilize a new backbone of diffusion models, Diffusion Transformer, for adaptive TTS. Specifically, the adaptive layer norm in the architecture was used to condition the diffusion network on text representations, which further enabled parameter-efficient adaptation. On basic TTS tasks, the new architecture was verified to be a faster alternative to its convolutional counterpart. For few-shot adaptation, the DiT decoder demonstrated a clear advantage on naturalness and speaker similarity over the Transformer decoder while maintaining parameter efficiency. When used in a zero-shot adaptive system, while we found the DiT is a decent alternative to the non-causal WaveNet, its main merit is to provide efficient high-quality adaptation when finetuning is performed.

## 5. Acknowledgments

This project received funding under NAST: Neural Architectures for Speech Technology, Swiss National Science Foundation grant 185010.

---

[5]https://github.com/Rongjiehuang/GenerSpeech

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
