# OpenReview forum: "Diffusion Transformer for Adaptive Text-to-Speech"
_Interspeech.org/2023/Workshop/SSW — SSW12_

### Official Review · Reviewer_imkd · 2023-06-04
**Diffusion Transformer for Adaptive Text-to-Speech**

**Rating:** 8
**Confidence:** 4

**Review:**

-Key Strength of the paper

Adaptative TTS is an important issue in order to provide high-quality and natural synthetic speech with requiring few data, few parameters, and low complexity.

The idea of this paper is to present a diffusion transformer as the conditionnal decoder of a neural TTS. This model is faster than conventional alternatives and have provided higher quality in the field of image generation.

Three experiments are proposed in order to evaluate the proposed model to three conditions : single-speaker synthesis, speaker adaptation with few-shot adaptation, and zero-shot adaptation.
Objective evaluation by means of cosine speaker similarity and character error rate are provided, and subjective evaluations by means of MOS for quality and naturalness are also provided.

The proposed model is compared to a so-called non-causal WaveNet for the single speaker task; and to the AdaSpeech algorithm for the speaker adaptation task.

-Main Weakness of the paper

The description section of the proposed model appears limited compared to the experimental section. More details and discussion should be provided to present the originality of the proposed model and its supposed advantages over other existing models.

In the experimental section, the comparison to the state-of-the-art is very limited.

The result of the third experiment suggests that the main adaptation is operated within the reference encoder which is not considered in this paper. This limits the interest for speaker adaptation.

More details about complexity could be provided. How is measured the real-time factor?


-Novelty/Originality, taking into account the relevance of the work for the SSW audience

Adaptive TTS is an important topic, and the proposed diffusion transformer inspired from image generation appears fully motivated for this task

-Technical Correctness, is the work technically and/or scientifically solid? Are sufficient details provided to allow any experiments to be reproduced or equivalent experiments run?

Yes, mostly

---

### Official Review · Reviewer_7Bia · 2023-06-05
**Adaptive transformer fo TTS**

**Rating:** 7
**Confidence:** 4

**Review:**

This paper proposes to adapt a diffusion transformer (DiT) to the case of adaptive TTS. The DiT model is then used as the decoder providing interesting benefits in terms of quality and speed.

First experiment compares the proposed DiT to a non-causal Wavenet and shows a similar output quality (naturalness) with a performance gain (70% faster compared to the wavenet baseline).
Second experiment investigates the adaptation ability of the DiT model. Results are really interesting, the improvement is clear. DiT show a good adaptation capacity using inly 10 utterances for the target speaker (improvement in the MOS and similarity evaluations).
The third experiment evaluates the DiT model in a 0-shot context. Results are more nuanced in that case.

Overall, the proposition is interesting and promising. It brings a concrete progress over the state of the art. I recommend to accept this paper.


Typos:
- "The nature cancels the requirement" etc.: I don't understand this sentence

---

### Decision · Program_Chairs · 2023-06-14

**Decision:**

Accept

**Comment:**

SSW2003 received 45 papers. The acceptance rate is 82%. We are pleased to inform you that your paper has been accepted by the SSW2023 Program Committee. Please read the reviews carefully and submit your camera-ready paper by June 28th. Most reviewers performed a detailed review. Please answer to their questions and consider their comments. Note that camera-ready papers are credited with one extra page to allow authors to consider reviewers’ suggestions. So max 7 pages in total including figures & refs.
The deadline for submitting the revised version (with full non-anonymized authors and refs!) is 28th June.